# Green Fabrication of Sustainable Porous Chitosan/Kaolin Composite Membranes Using Polyethylene Glycol as a Porogen: Membrane Morphology and Properties

**DOI:** 10.3390/membranes13040378

**Published:** 2023-03-26

**Authors:** Sonia Bouzid Rekik, Sana Gassara, André Deratani

**Affiliations:** 1Institut Européen des membranes, IEM, UMR-5635, ENSCM, CNRS, Université Montpellier, 34095 Montpellier, France; 2Bioengineering, Tissues and Neuroplasticity, EA 7377, Faculté de Santé, EPISEN, Université Paris-Est Créteil, 8 rue du Général Sarrail, 94010 Créteil, France

**Keywords:** biopolymer/inorganic mixed matrix, phase inversion, pore-former additive, molecular weight effect, asymmetric morphology

## Abstract

One of the major challenges in membrane manufacturing today is to reduce the environmental footprint by promoting biobased raw materials and limiting the use of toxic solvents. In this context, environmentally friendly chitosan/kaolin composite membranes, prepared using phase separation in water induced by a pH gradient, have been developed. Polyethylene glycol (PEG) with a molar mass ranging from 400 to 10,000 g·mol^−1^ was used as a pore forming agent. The addition of PEG to the dope solution strongly modified the morphology and properties of the formed membranes. These results indicated that PEG migration induced the formation of a network of channels promoting the penetration of the non-solvent during the phase separation process, resulting in an increase in porosity and the formation of a finger-like structure surmounted by a denser structure of interconnected pores of 50–70 nm in diameter. The hydrophilicity of the membrane surface increased likely related to PEG trapping in the composite matrix. Both phenomena were more marked as the PEG polymer chain was longer, resulting in a threefold improvement in filtration properties.

## 1. Introduction

Membrane filtration systems are widely used in separation and purification processes, including wastewater treatment and drinking water production [1,2,3]. Compared to other materials, polymer-based membranes, the most commonly used membranes in water treatment, have the advantages of being inexpensive, and easy to process and scale up [4]. Most often, they consist of polymers derived from oil, such as polysulfone (PSf), polyethersulfone, polyvinylidene difluoride (PVDF), and so on. The general trend towards sustainability requires a paradigm shift in the membrane preparation industry to explore and develop the use of green processes and of renewable and non-hazardous chemicals [5]. For example, the use of biocompatible and biodegradable materials in membrane matrices can serve as an important criterion to be fulfilled to produce eco-friendly membranes [6,7,8]. In this regard, many polysaccharides have film-forming properties that make them suitable for the preparation of membranes with different characteristics [9].

Phase inversion from homogeneous polymer solutions to a solid membrane, is the most widely used fabrication technique [10]. However, the phase inversion process requires the use of organic solvents for polymer solubilization. The new environmental concerns necessitate evaluation of the impact of the toxicity of these solvents on the environment but also on the workers’ health. Regulatory constraints and increasing prices of raw materials have a significant impact on the cost of the membrane manufacturing process. Therefore, sustainable alternatives, provided they are economically feasible and environmentally beneficial, are also an industrial concern, and any changes in the process that can lead to a cleaner technology would be of great value [11,12].

Faced with the challenges of reducing these impacts, many efforts have been devoted to the replacement of conventional hazardous solvents with more environmentally friendly solvents including ionic liquids and bio-based solvents [5,13]. However, to date, the actual environmental and health impact of these new solvents, including the entire life cycle from their fabrication to the amount and toxicity of waste generated, has not yet been truly addressed [11]. On the other hand, water seems to be an ideal candidate from both the general impact and economic point of view. Over the past decade, our team has developed the fabrication of non-toxic and biodegradable polymer-based membranes using water as a solvent and non-solvent. The decrease in polymer solubility, induced by an increase in temperature above the lower critical solution temperature, promoted the phase separation of poly(vinyl alcohol) [14] and of hydroxypropyl cellulose [7]. Maintaining the membrane morphology and porous structure at room temperature required post-treatment with chemical crosslinkers, as these polymers are water-soluble. However, there are some constraints on the use of the resulting membranes because they are sensitive to the pressure applied during filtration due to the high water swelling of the hydrophilic polymer matrices.

Addition of a mineral filler as a reinforcing additive to stabilize the membrane structure can overcome this problem. This approach requires high compatibility and close interaction between the mineral particles and the polymer chains. Chitosan (CS), a cationic polysaccharide, was chosen for this study due to its ability to establish ionic bonds through electrostatic interactions with negatively charged particles. CS, obtained from chitin of crustacean shells, which is the second most common natural polymer after cellulose, is an attractive choice because it represents a good balance between human food consumption and valorization of industrial by-products [15]. In addition, CS is a very popular material owing to its excellent film-forming ability, hydrophilicity and ease of modification through its reactive amine groups [9]. These properties make it a well-suited biomaterial in the preparation of membranes for a variety of applications including gas separation, pervaporation, purification processes, reverse osmosis, drug delivery and microbial fuel cells [15,16,17]. However, pure CS membranes show some shortcomings in terms of mechanical strength, thermal stability and solubility in acidic media which generally limit its application in dense membranes in aqueous–organic media [16,18].

During our study on sustainable membrane fabrication, a new environmentally friendly method was explored using a pH gradient, as CS is only soluble in water under acidic conditions due to the ionization of amino groups (pKa about 6). Composite membranes were developed based on CS reinforced with negatively charged kaolin (KO) particles [19,20]. The resulting membranes exhibited drastic changes in their properties compared to the corresponding pure CS material, such as improved mechanical and thermal properties, chemical stability, etc., highlighting the inorganic counterpart as a reinforcement for the polymer matrix. A crosslinking post-treatment even improved the stability in acidic media up to pH 2 [21]. A stable porous membrane series with good water permeability and pore size in the range of microfiltration and ultrafiltration, depending on the preparation conditions, was thus obtained.

Porosity in the CS/KO composite resulted predominantly from the introduction of water-soluble polyethylene glycol (PEG) into the dope solution. PEG and polyvinylpyrrolidone are the most common pore-forming macromolecules used to generate high permeate flow. Porosity is created by leaching out most of the additive while the remaining part entrapped in the membrane matrix contributes to the enhancement of membrane surface hydrophilicity [22,23,24]. Many works have reported on the role played by PEG on the membrane morphology and permeability as a function of its content and molecular weight (Mw), for example in the case of PSf [22,25], polyphenylsulfone [26] and PVDF [23]. The general trend is that the membrane performance varied greatly with the Mw of PEG, provided a minimum amount (6–10%) was added. On the other hand, the optimal Mw according to the literature depended on the polymer matrix and the membrane formation conditions.

Therefore, it seems obvious that formulation optimization of the dope solution used to prepare the CS/KO composite membranes should analyze the influence of the Mw of PEG. The present work focused on the impact of four different PEG Mw (400, 2000, 6000 and 10,000 g·mol^−1^) as an additive on the membrane formation mechanism, morphology, porosity and hydrophilic–hydrophobic balance. The rheological properties of the dope mixtures were investigated with respect to the effects of shear rate and temperature on the dynamic viscosities. The thermal and chemical stability of the prepared membranes were also determined as well as the filtration properties in terms of water flow rate and permeability.

## 2. Material and Methods

### 2.1. Chemicals

Chitosan (CS), with a mass average molecular weight of 180,000 g·mol^−1^ and a degree of deacetylation of 80% was obtained from France Chitine (Orange, France). The clay used in the present study was a kaolin Codex (noted as KO) provided by the Laboratoire des Plantes Médicinales (Tunis, Tunisia). Polyethylene glycols of Mw 400 g·mol^−1^ (PEG-400) and 10,000 g·mol^−1^ (PEG-10000) supplied by Fluka (Seelze, Germany), and Mw 2000 g·mol^−1^ (PEG-2000) and Mw 6000 g·mol^−1^ (PEG-6000) supplied by Sigma-Aldrich (St. Louis, MO, USA), were used for the preparation of casting solutions. Analytical grade sodium tripolyphosphate (STPP) crosslinking agent was obtained from Sigma-Aldrich. Acetic acid, hydrochloric acid and sodium hydroxide pellets were laboratory grade chemicals. All aqueous solutions were prepared using deionized water (18 MΩ/cm, Millipore Milli-Q; Fontenay sous bois, France).

### 2.2. Composite Membrane Preparation

The pure CS solution and the composite suspensions were prepared following a procedure previously described [19]. First, CS powder (4 *w*/*v*% in 0.1M acetic acid) was allowed to dissolve under stirring at room temperature. Then, the desired amount of KO powder was added slowly under stirring to obtain a homogeneous suspension. The casting suspension (named CKPEG-n, n standing for the PEG Mw) was finally obtained by dissolving the PEG powder at a content of 1 *w*/*v*% under stirring. Table 1 summarizes the composition of the different casting suspensions prepared. The resulting suspensions were degassed for 24 h under vacuum to remove air bubbles and then cast onto a glass plate with a 700 μm gap casting knife using an automatic coater (K Control coater, Erichsen; Hemer, Germany). The forming membrane was dried at room temperature to partially evaporate the solvents and immersed into a 1 M NaOH solution for 24 h [27]. After rinsing the membrane thoroughly with deionized water to remove excess NaOH and reach a neutral pH value, post-treatment was performed by soaking in a solution of STPP (5 *w*/*v*%) for 20 min at 25 °C and pH = 4 (by acidification with HCl) [28]. The resulting crosslinked membranes were thoroughly washed with DI water to remove excess crosslinking agent and stored in DI water [21].

### 2.3. Rheological Properties of Casting Mixtures

The rheological behavior of different casting mixtures was measured with an Anton Paar MCR301 rheometer (Les Ullis, France), using a 50 mm plate geometry as measuring tool. The influence of temperature on the dynamic viscosity of mixtures was also determined. The temperature scan tests were performed in the range of 20 °C to 80 °C.

### 2.4. Membrane Characterization

#### 2.4.1. Membrane Morphology and Pore Size Determination

The surface and cross-sectional morphology of the obtained membranes was examined using scanning electron microscopy (SEM) (Hitachi S-4500, 1.5 nm resolution at 15 kV; Meudon La Forêt, France). The samples were dried at room temperature, cut into small pieces, and coated with a thin layer of Pt by sputtering before SEM analysis. For cross-sectional image acquisition, the samples were first immersed in liquid nitrogen before being broken up. The average pore size and pore size distribution were estimated from 5 to 8 images taken from samples of at least three different membranes by processing the surface SEM images with ImageJ analysis freeware (https://imagej.nih.gov/ij/, ImageJ-Win64 downloaded 22 September 2019). Nitrogen adsorption–desorption isotherms at 77 K (Micromeritics ASAP 2020; Norcross, GA, USA) were applied for determination of the mean pore diameter and pore volume.

#### 2.4.2. Mechanical Property Measurements

The membrane mechanical properties were evaluated using extensional rheology. These tests were performed with an MCR 301 rheometer (Anton Paar) using a universal extensional fixture UXF12. The temperature was controlled at 25 °C with a CTD180 Peltier system. Pieces of 4 × 1 cm were cut from different parts of the membranes (at least 3 per formulation). This test was performed on samples in the wet state. The wet state corresponds to a membrane stored in water for more than 24 h.

#### 2.4.3. Thermal Stability

Thermogravimetric analysis was performed using a Hi-Res TGA 2950 device (TA instrument, Guyancourt, France). Experiments performed on approximately 6 mg of the composite membranes were scanned from 25 to 800 °C at a heating rate of 10 °C/min under a nitrogen atmosphere to prevent possible thermo-oxidative degradation.

#### 2.4.4. Water Solubility

The membranes’ solubility in water (natural pH), acidic (pH = 4) and basic (pH = 9) media was determined from the weight loss of dry samples. It was defined by the solubilized dry matter content after 24 h of immersion in the considered medium. Each membrane sample (20 mm × 20 mm) was dried in an oven at 80 °C to a constant weight and then immersed in 50 mL of the given medium. After a 24 h contact at room temperature, the membrane pieces were removed and dried to a constant weight in an oven at 80 °C to determine the weight of insolubilized dry matter. The measurement of the soluble fraction was determined as follows:(1)$$SOL=\frac{M_{i}-M_{f}}{M_{i}}\times 100,$$
where *SOL* is the percentage of soluble matter, and *M_i_* and *M_f_* are the initial and final sample mass, respectively.

#### 2.4.5. Water Contact Angle (WCA) Measurement

Membranes were washed with Milli-Q water, dried on a paper filter overnight, and stored in a desiccator until WCA measurement. Droplets of approximately 10 µL of DI water were deposited on the membrane surface using a microsyringe. The contour of the droplet was analyzed using ImageJ analysis freeware. The contact angle value was then determined using interpolation methods (DropSnake). To minimize the experimental error, the contact angle was measured at least at three random positions and their average values are reported.

#### 2.4.6. Infrared Spectra Measurements

Infrared spectra of PEG powder and composite membranes were recorded using a Nicolet 710 Fourier-transform infrared (FTIR) spectrometer with attenuated total reflection (ATR). The spectra were acquired in the wavenumber range from 4000 to 650 cm^−1^ with 64 scans at a 4 cm^−1^ resolution.

#### 2.4.7. Filtration Tests

Frontal filtration experiments were performed using an agitated dead cell (Amicon 8050, Millipore Corporation; Fontenay sous bois, France) with membranes having an active surface area of 13.4 cm^2^ according to a previously described procedure [29]. The membranes were first conditioned in the test cell with pure water by gradually increasing the pressure to 4 bar for at least 3 h. Then, the cell was connected to a water tank and pressurized to the desired value using compressed air. The mass of permeate passing through the membrane was recorded at regular intervals using SartoConnect software (Sartorius, Göttingen, Germany). Pure water flow rate (*PWF*) was measured for each membrane by flowing DI water through the membrane system with an applied pressure between 0.5 and 3.5 bar. *PWF* (L·m^−2^·h^−1^) was calculated using the following formula,
(2)$$PWF=\frac{Q}{∆t\times A},$$
where *Q* (L) is the volume of permeate, Δ*t* (h) the permeation time and *A* (m^2^) the active membrane area. In each experiment, *PWF* increased linearly with operating pressure. Pure water permeability *PWP* (L·m^−2^·h^−1^·bar^−1^) was determined from the slope of the linear variation of *PWF* with applied pressure.

## 3. Results and Discussion

### 3.1. Rheological Properties of Casting Mixtures

The rheology of casting mixtures is an important parameter that controls morphology and helps to achieve uniform thickness and defect-free membranes [30]. Figure 1 shows the variation of viscosity with temperature changes from 20 °C to 80 °C. As can be seen, the addition of KO to the CS solution led to an initial increase in the viscosity of the resulting suspension. This behavior accounts for electrostatic interactions and hydrogen bonding that can take place due to the existence of amino and hydroxyl groups on CS, and negatively charged and hydroxyl groups on the mineral particles [19]. The additional increase in viscosity observed upon addition of PEG indicates that good mixing took place with the CS/KO suspension resulting in increased mutual interactions between the particles. It can be assumed that the electron donating groups of PEG can create strong hydrogen bonds with the hydroxyl and amine groups, leading to the insertion of PEG chains into the network formed between the CS and clay layers. The viscosity of the casting suspensions became higher as the PEG chains were longer, as expected from the literature [31]. An increase in temperature caused a sharp decrease in the suspension viscosity due to chain relaxation. In the casting procedure, the temperature was set to 25 °C under conditions where the viscosities of the suspensions were high enough to give defect-free membranes. Further, Figure 2 shows that the composite mixtures exhibited a Newtonian behavior, meaning that the viscosity was essentially independent of the shear rate in the range used (about 95 s^−1^).

### 3.2. Membrane Morphology

KO particles have a negative surface charge in the pH range 3–9 (isoelectric point about 3). CS is positively charged in the same pH range (pKa = 6.5). Mixing the two leads to strong interactions through charge neutralization of KO particles and hydrogen bonding [19] so that the zeta potential shifts to less negative values depending on the CS/KO ratio and the CS Mw [32]. The zeta potential of CS/KO suspensions prepared at pH 4 in this work was positive and equal to about 15–20 mV. It has been reported that PEG can also adsorb to KO particles and cause an increase in zeta potential although these molecules are neutral [33]. However, due to the large difference in concentration, Mw and interaction strength compared to CS, PEG probably decorated the CS/KO surface as no significant difference in zeta potential was observed. Upon neutralization of the film forming system, precipitation–flocculation occurred. It was expected that PEG would be released into the coagulation bath and act as a pore former.

SEM analysis is a powerful technique to examine the structure of membranes and to obtain significant information regarding surface morphology and cross sections. Figure 3 compares SEM images of cross sections of the prepared membranes. The cross section of the CS membrane (Figure 3a) showed a dense structure without any holes while that of the CK (Figure 3b) membrane had a loosely layered structure that was assumed to be due to the intercalation of CS chains between KO sheets [19]. Only a slight change in morphology after PEG-400 incorporation was visible in SEM images of the CKPEG-400 membrane (Figure 3c). In contrast, the addition of PEG with Mw of 2000, 6000 and 10,000 g mol^−1^ resulted in an increase in porosity with the appearance of finger-like structures under a sponge-like structure at the surface (Figure 3d–f). This type of asymmetric morphology is usually observed in the case of rapid phase de-mixing during the membrane formation process [34,35]. Once immersed in the coagulation bath, the CS/KO system experienced fast proton exchange leading to solidification of a CS network with KO particles immobilized within. At the same time, the highly water-soluble PEG migrated to the less concentrated medium, creating the porous interlacing at the interface. The sponge-like top layer formed delayed the exchange between the membrane-forming system and the coagulation bath. Convection instabilities had time to establish, generating large variations of local polymer concentration, resulting in macrovoid structures. PEG probably diffused very easily within the macrovoids consisting mainly of water, and then to the coagulation bath, forming parallel channels (finger-like structure). These findings are consistent with the membrane structures obtained from other polymer solutions and mineral particle suspensions using PEG as a pore-former [31,36].

As seen in the SEM pictures (Figure 3d–f)), the length of the finger-like structures increased with increasing PEG Mw. Furthermore, the number of finger-like pores became larger for PEG Mw above 2000 g∙mol^−1^. Specifically, the CKPEG-10000 composite membranes, prepared with PEG of Mw 10,000 g∙mol^−1^ as an additive, exhibited the greatest number of finger-like pores, and the longest ones. These structures extended almost entirely across the membrane in a parallel arrangement, allowing good interconnection between the feed and permeate sides. As explained above, the hydrophilic nature of PEG promoted the diffusion of water during the phase inversion process, which resulted in a substantial increase in pore formation. The reason for the observed differences in morphology between the CKPEG-400 and CKPEG-10000 membranes was probably the change in the role of the additive in the casting solution when going from the PEG-400 oligomer to the PEG-10000 macromolecule. In the first case, PEG-400 can be considered as a solvent molecule due to its small size and highly hydrated state. Therefore, the morphology of CKPEG-400 and CK membranes was almost identical, and no significant effect of the additive could be detected, except for a slight increase in the membrane thickness. In contrast, the CS/PEG-10000 system behaves more like a mixture of two polymers in a solvent/non solvent medium [26]. PEG additives of Mw 6000 and 2000 g·mol^−1^ showed intermediate behavior, leading to a morphology of CKPEG-6000 and CKPEG-2000 membranes similar to that of CKPEG-10000 and CKPEG-400, respectively.

Thus, it appeared that the development of finger-like structures became more pronounced with the higher Mw PEGs as additives, probably due to their hydrophilicity and low compatibility with CS in the solid state [37]. On the other hand, an increase in PEG Mw led to a substantial increase in the viscosity of the composite suspension (Figure 1). A higher viscosity was expected to reduce the exchange between the membrane forming system and the coagulation bath, promoting the development of a thicker sponge-like structure at the interface [22,31]. This is demonstrated in Figure 4, by the steady evolution of the average thickness of the sponge-like surface layer determined from SEM images (Figure 3c–f) with increasing PEG Mw. The total membrane thickness similarly increased, as shown in Figure 4, to about 57 and 116 µm for CKPEG-400 and CKPEG-10000 membranes, respectively, indicating a large increase in sublayer porosity. As mentioned above, low Mw PEG was more easily removed from the CS/KO system during the phase inversion than a high Mw PEG. Therefore, the latter additive tended to be more concentrated in the forming pores resulting in larger finger-like structures [31].

### 3.3. Surface Porosity, Average Pore Size and Pore Size Distribution

Figure 5 shows the evolution of the surface morphology of the pure chitosan (CS) film and the CK composite membrane. The CS films had a continuous structure with a smooth, homogeneous and compact surface without pores or cracks, while the CK composite membrane had a much rougher surface morphology. A zoomed-in image clearly reveals the presence of tiny pores inside the particles, whose size can be estimated as between 20 and 60 nm using image analysis. The origin of the pore structure was assumed to be the migration of chitosan molecules to the negative charges between the clay sheets. Thus, the penetration and intercalation of chitosan chains into the clay layers would result in the formation of a three-dimensional network of nano-sized pores in the composite membrane [19].

Figure 5 also shows surface SEM images of the membranes prepared with varying PEG Mw to compare the obtained pore size and density. As expected, the presence of PEG additive in the casting suspension resulted in a significant pore opening and pore size increase compared to the CK composite membrane. In addition, the increase in PEG Mw caused a large increase in surface porosity. SEM image analysis of the membrane surface was used to quantify these parameters (Table 2) [29,31]. The average surface pore diameter increased slightly with increasing PEG Mw to about 50 and 70 nm in the case of CKPEG-400 and CKPEG-6000 membranes, respectively, while the surface porosity showed a threefold increase. Interestingly, the variation leveled off for higher PEG Mw. The surface porosity and average pore size values did not seem to be strongly affected, leading to very similar values for the Mw 6000 and 10,000 g∙mol^−1^. These results may indicate that these two additives behaved identically at the interface during phase inversion. The trend of increasing average surface pore size and pore density was attributed to the leaching of PEG from the casting solution. It was also assumed that the hydrophilic contribution of PEG creates interspaces between the polymer chains and the clay layers, thereby increasing the porosity of the composite matrix. When the Mw of PEG is higher than 6000 gmol^−1^, the effect on the average pore size and pore density can be counterbalanced by the opposite influence related to the increased in viscosity involving reduced mobility of polymer chains [31,38]. The data obtained from SEM image analysis are compared in Table 2 with the pore size and volume determined using nitrogen sorption–desorption measurements. The pore size was systemically smaller and the pore volume showed an opposite trend to those observed in the SEM images (Figure 3 and Figure 5). The procedure used in the method probably led to a partial collapse of the membrane structure, with a stronger collapse the larger the PEG Mw, and, hence, the higher the porosity. Therefore, the SEM image analysis appeared to reflect the morphology of CS/KO composite membranes more accurately.

The pore size distribution (PSD) of the composite membranes was determined as a function of PEG Mw using surface SEM image analysis (Figure 6). As can be seen, the PSD broadened with increasing PEG Mw, with CKPEG-400 showing the narrowest PSD. The longer the PEG chains, the more easily they can entangle with the CS/KO network due to higher viscosity and lower mobility during the phase inversion process. This effect was assumed to contribute significantly to the broadening of the PSD with increasing PEG Mw.

### 3.4. Surface Properties of CS/KO Membranes

The surface of the membrane, which is in direct contact with the feed solution, plays a fundamental role in separation properties. Surface chemistry and wettability of the prepared CS/KO membranes were characterized using ATR–FTIR and WCA measurements.

Figure 7 presents the FTIR spectra of the CKPEG-10000 membrane surface compared to that of PEG-10000 powder and CK membrane. The CK spectrum showed the presence of KO by its very strong bands at 996, 1025 and 1112 cm^−1^, ascribed to the stretching vibrations of Si–O, and that of CS by bands at 1650 and 1590 cm^−1^, attributed to C = O stretching vibration (amide I band) and N–H stretching vibration (amino II band), respectively. The interaction between CS and KO was evidenced by a shift of these characteristic bands compared to the starting materials [39]. This effect confirmed the ionic exchange reaction occurring between CS and kaolin-clays and intercalation into the KO structure [19]. PEG was characterized by two broad bands at about 2974–2881 and 1148–1060 cm^−1^, which correspond to the absorption of aliphatic C–H and ether C–O–C groups, respectively [40]. Using PEG as an additive in the CK composite membrane, notable changes appeared in the 1150–950 cm^−1^ range, which can be explained by the signature of the PEG ether groups. This result confirms the presence of trapped PEG chains at the membrane surface.

Hydrophilicity is a desirable property for filtration membranes in aqueous media because it is expected to give high permeate flow rates and reduce fouling. Figure 8 shows the results of WCA measurements for the different composite membranes prepared. Wettability is a function of surface chemistry but also of surface roughness [40]. The pure CS membrane showed a WCA value of about 57°, before the water droplet was fully absorbed by the matrix. This result reflects the marked hydrophilic character of the CS polysaccharide backbone and the smooth membrane surface, as observed using SEM (Figure 5). It can be clearly seen that the surface morphology became rougher after KO loading compared to that of the pure CS membrane. In this case, the surface roughness played a key role in imparting an apparent hydrophobic character to the membrane surface. Aggregated particles could be observed uniformly distributed on the CK membrane surface, inducing a high degree of surface roughness explaining the higher WCA value. On the other hand, the H and ionic bonds between the functional groups of CS and KO generated a stiffer membrane surface and reduced interactions of hydrophilic groups with water molecules, which also participated in the increase in the WCA value [19]. In contrast, the incorporation of PEG caused drastic changes in the hydrophilic properties of the resulting membranes. The PEG addition induced a smoothening effect on the membrane surface, which was particularly noticeable in the case of CKPEG-10000. The WCA of the CKPEG-10000 showed a value as low as about 62° (Figure 8), indicating a significant increase in wettability with respect to the CK membrane surface. Further, ATR–FTIR analysis showed that a part of PEG chains remained entangled in the CS/KO network, leading to a proportion of residual PEG on the membrane surface. Therefore, it was assumed that the highly hydrophilic nature of the PEG chains located at the interface resulted in an increase in wettability through greater affinity of the membrane surface for water molecules [31,41].

### 3.5. Physical Properties of the Composite Membranes

Mechanical properties are critical parameters for the applicability of membranes in pressure-driven filtration processes. The tensile strength and elongation at break of the prepared composite membranes were measured in wet state (Figure 9 and Table 3). The results show that they were directly dependent on the increase in PEG Mw in the casting solution. However, as seen previously, PEG Mw strongly influenced the structure of the membrane: long chains accentuated the asymmetric character of the morphology with a dense and sponge-like surface layer on top of a sub-layer mainly made of finger-like macrovoids. In fact, the dense surface layer provided the majority of the mechanical strength of the asymmetric membranes. Figure 4 shows that this layer became thicker with the increase in the Mw of PEG which is in complete agreement with the increase in tensile strength observed in Table 3. Moreover, the porosity of the membranes also increased strongly due to the creation of free volumes and finger-like pores related to the release of PEG molecules into the coagulation bath. The reduction in the elongation at break when comparing the CKPEG-400 and CKPEG-10000 membranes was a result of this phenomenon. The addition of PEGs with higher Mw as pore formers therefore resulted in two opposite effects on the mechanical properties: the enhancement of the tensile strength by a thickening of the dense surface layer and a decrease in the plasticity (elongation at break) due to the formation of macrovoids.

The thermal stability of the prepared composite membranes was determined using TGA (Figure 10). Three independent regions can be observed. The first weight loss (50–150 °C) was attributed to the release of the more or less adsorbed water molecules in the composite membrane, the second (150–450 °C) to the partial decomposition of CS and the remaining PEG chains, while the third (400–600 °C) was the result of the simultaneous dehydroxylation of kaolinite and final degradation of the CS chains [20]. The Mw of the PEG used did not significantly affect the thermal stability of the resulting membranes. On the other hand, the increase in Mw led to a higher amount of adsorbed water, which is in agreement with the results of increased porosity and hydrophilicity in the membranes mentioned in the previous sections.

Resistance to water washout is an important parameter to consider when using CS-containing membranes in water treatment applications as protonation of the amine groups in acidic media leads to its partial dissolution. Therefore, the water solubilization (WS) of the pristine CS membrane and membranes prepared with different PEG Mw was studied under different pH conditions (pH = 2, 4, 9, and at the natural pH of DI water (6.2)) to investigate their chemical stability in aqueous media (Table 4). As expected, the CS membrane dissolved at pH values below neutrality. In fact, incorporation of clay particles through strong electrical interactions and the CS crosslinking prevented acidic deterioration of the membrane [19,20,21]. Interestingly, no negative effects on chemical stability were observed using PEG of varying Mw and the composite membranes obtained were stable down to pH 2.

### 3.6. Membrane Permeability

Figure 11 shows the pure water permeability (PWP) obtained from the slope of the linear variation of the pure water flow with the applied pressure (0–3.5 bar) for the different prepared membranes. The CS membrane exhibited very poor permeability due to its dense structure and lack of porosity (Figure 3 and Figure 5). The addition of KO particles led to an improvement of one order of magnitude in PWP, which is consistent with the porosity revealed in SEM images (Figure 5). An increase in PEG chain length from 400 to 10,000 g∙mol^−1^ resulted in a threefold enhancement of PWP. Thus, these results are in good agreement with those presented in the previous sections in terms of membrane porosity, pore size and hydrophilicity. However, the thickness of the top layer had the opposite effect due to the pore tortuosity that impeded permeate flow. It was then assumed, that the finger-like pores extending over almost the entire CKPEG-10000 membrane, as well as its hydrophilicity probably accounted for the highest permeability observed despite its thicker sponge-like top layer compared to the other composite membranes (Figure 4 and Figure 5). In contrast, the low Mw PEG additive did not lead to high enough porosity to improve PWP compared to the CK membrane. These results agree with poor PWP reported in the literature when using PEG additives with a low Mw [22].

As a conclusion, the characteristics of CS/KO composite membranes in terms of pore size and contact angle should lead to better permeability. However, membrane permeability depends on the surface wettability and pore size, but also on other factors such as filtration layer thickness, surface pore density and pore tortuosity. For example, the CKPEG-10000 membrane had a top filtration layer of about 25 µm thickness that strongly hindered the permeate flow (Figure 3f and Figure 4). It is obvious that a thinner top layer would result in better permeability. This point should be optimized to obtain more efficient membranes.

Many works have investigated CS-containing composites for various applications such as membrane filtration, metal ions, and adsorption of dyes and pollutants [42]. For comparison, Table 5 summarizes ceramic and polymer composites prepared according to several methods, including blending (as in this work), coating or layer-by-layer deposition on a pre-formed membrane. CS was immobilized using either chemical crosslinking with glutaraldehyde (GA) or hydrogen bonding and ionic interactions with negatively charged components (ceramic and carbon derivatives). Although the filtration range varies from MF to NF, it can be seen that the filtration performance of the CS/KO composite membranes prepared in this work using PEG Mw of 10,000 g∙mol^−1^ compares favorably with that of other works. This is probably due to their asymmetric morphology with a relatively thin filtration layer on a substructure providing very low permeate flow resistance. 

## 4. Conclusions

CS/KO composite membranes were prepared using PEG with Mw ranging from 400 to 10,000 g·mol^−1^ as a pore forming additives. The obtained membranes showed strong changes in their morphology with increasing PEG Mw, evolving from a homogeneous structure to an asymmetric one with a highly porous sublayer surmounted by a sponge-like top layer. At the same time, surface porosity and hydrophilicity were increased, indicating that PEG not only acted as a pore former but also as hydrophilic modifier due to chain entrapment during phase inversion. All these effects led to a strong increase in membrane permeability, with the best membrane prepared using PEG with a Mw of 10,000 g·mol^−1^. On the other hand, the PEG addition did not alter the physical properties (mechanical strength and resistance to acidic pH) so that these composite membranes could be a promising alternative to address the sustainability challenge in membrane fabrication for water treatment. Future work will investigate the potential of these membranes for the filtration of real feed solutions.

## Figures and Tables

**Figure 1 membranes-13-00378-f001:**
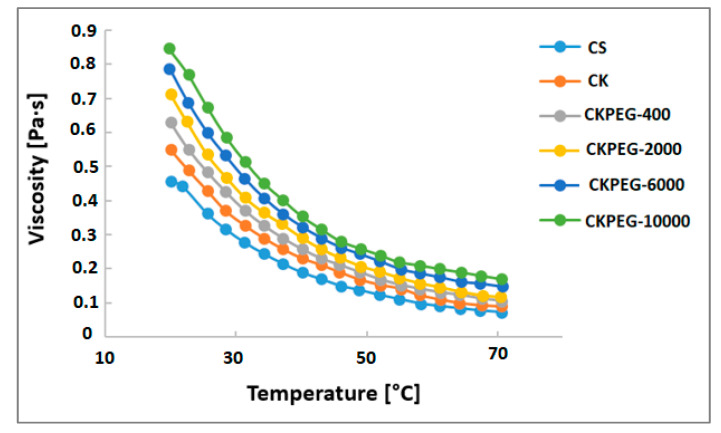
Evolution of the viscosity with temperature for the different prepared dope solutions.

**Figure 2 membranes-13-00378-f002:**
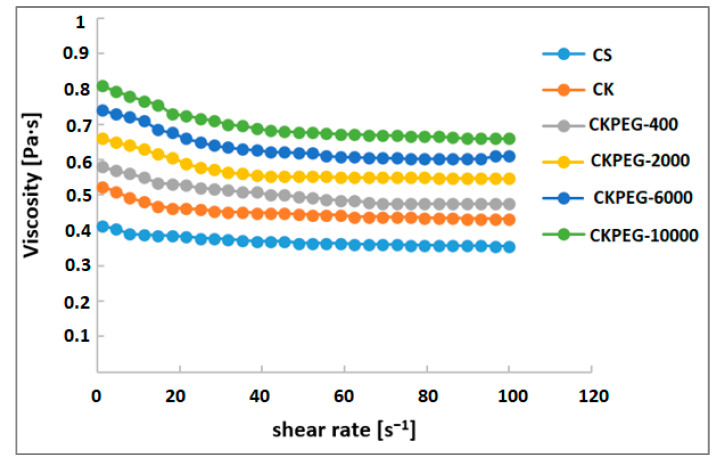
Influence of the shear rate on the rheological curves of different composite suspensions (T = 25 °C).

**Figure 3 membranes-13-00378-f003:**
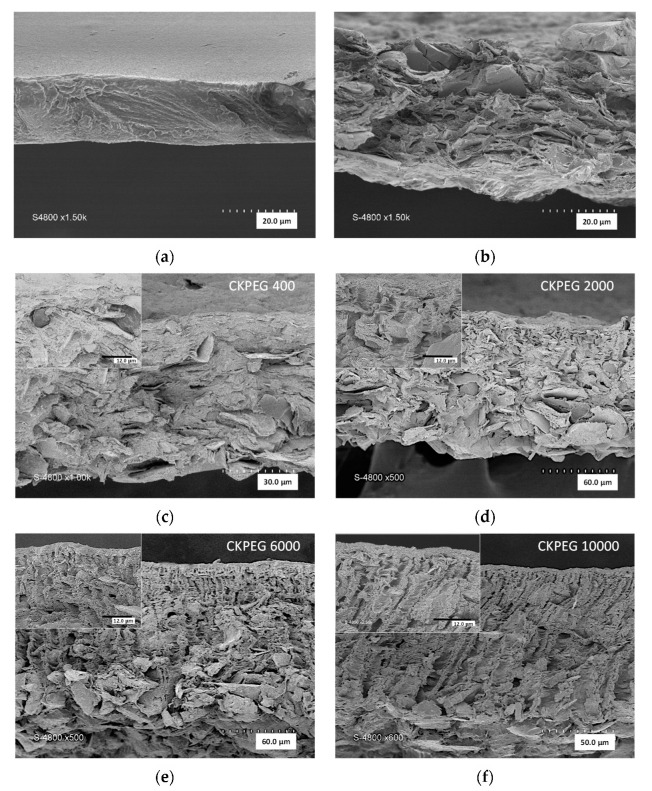
SEM Cross-sectional images of membranes prepared from: (**a**) pure CS membrane (×1500); (**b**) CS/KO (×1500); (**c**) CS/KO/PEG-400 (×1000); (**d**) CS/KO/PEG-2000 (×500); (**e**) CS/KO/PEG-6000 (×500); (**f**) CS/KO/PEG-10000 (×600). Insets are magnifications showing the sponge-like top layer. Magnification, ×2500; scale bar, 12 µm.

**Figure 4 membranes-13-00378-f004:**
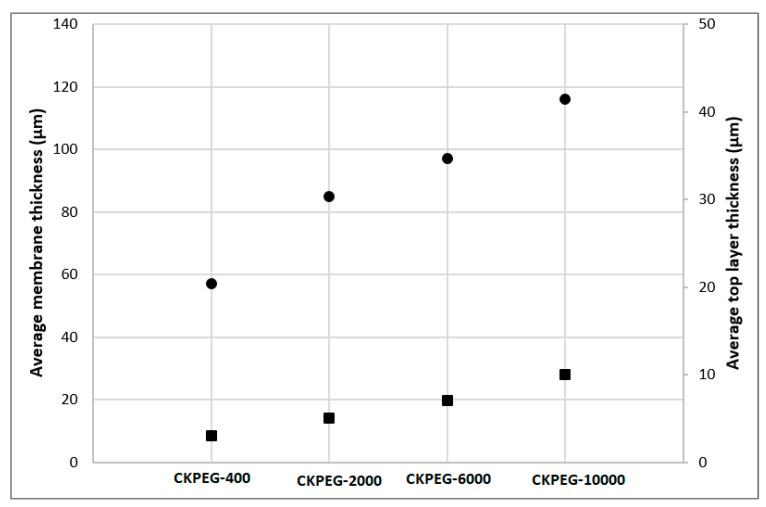
Variation of the average total membrane (●) and sponge-like top layer (■) thickness with PEG Mw.

**Figure 5 membranes-13-00378-f005:**
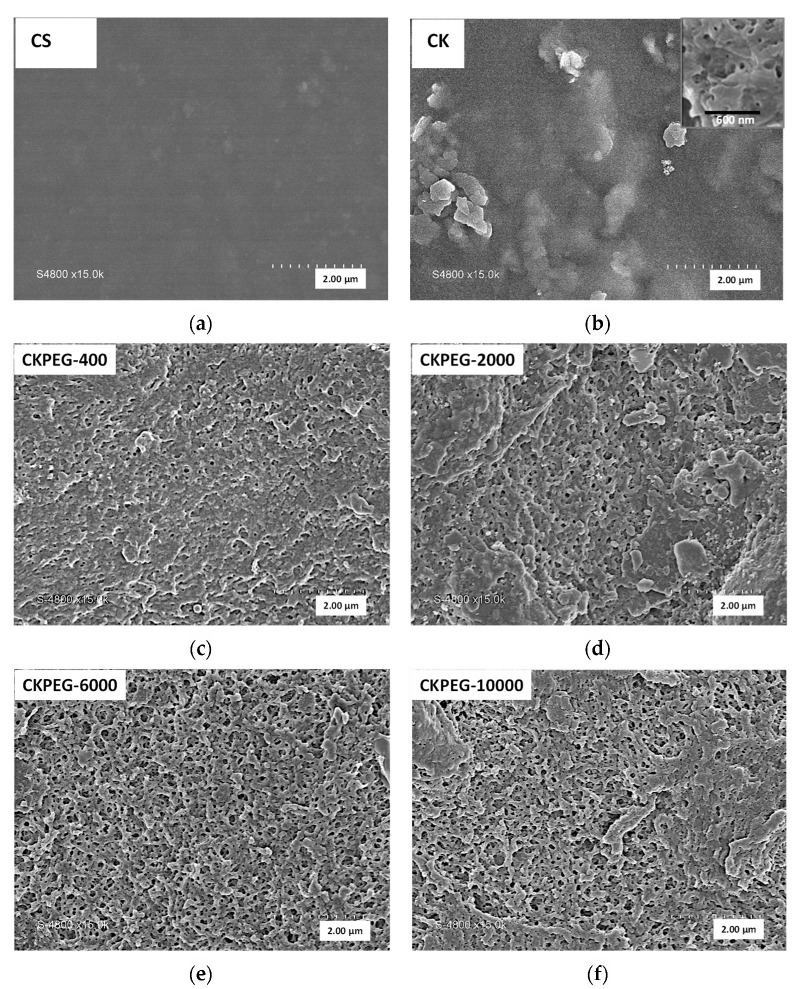
SEM surface images (magnification ×15,000) of a pure CS membrane (CS); a CS/KO composite membrane (CK) and CS/KO composite membranes with varying PEG Mw: 400 g·mol^−1^ (CKPEG-400); 2000 g·mol^−1^ (CKPEG-2000); 6000 g·mol^−1^ (CKPEG-6000); 10,000 g·mol^−1^ (CKPEG-10000). Inset is a magnified view showing the porous structure of KO particles in the CK membrane. Magnification, ×50,000; scale bar, 600 nm.

**Figure 6 membranes-13-00378-f006:**
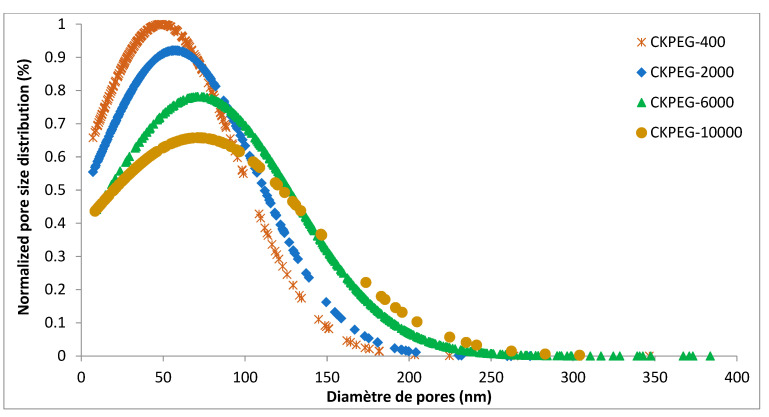
Pore size distribution determined using image analysis of SEM images depicting the surface of composite membranes prepared with varying PEG Mw as pore formers.

**Figure 7 membranes-13-00378-f007:**
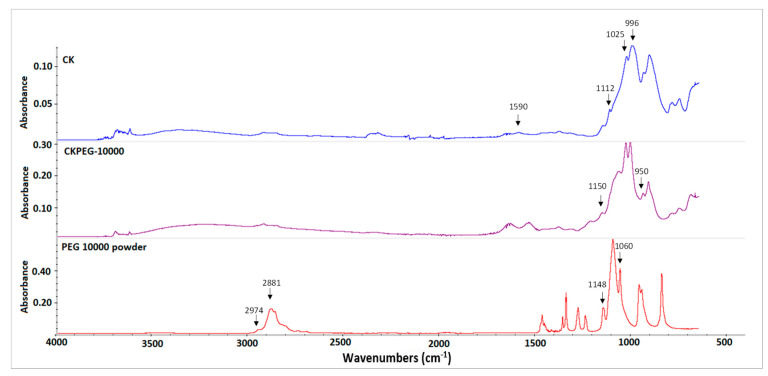
ATR–FTIR spectra of PEG powder, and CK and CKPEG-10000 composite membranes.

**Figure 8 membranes-13-00378-f008:**
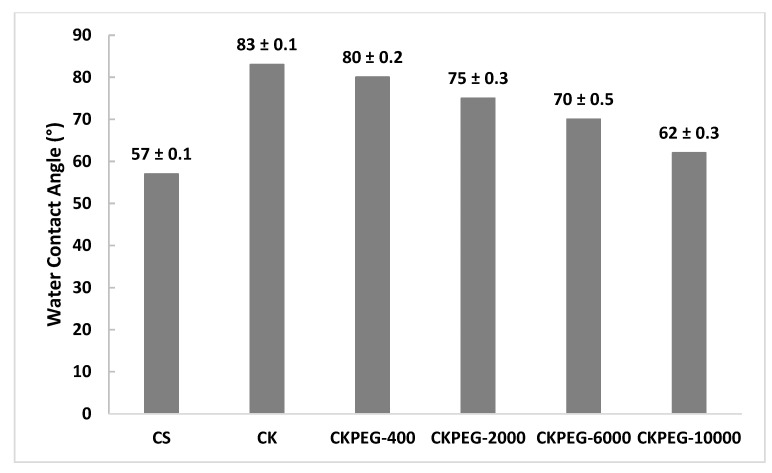
Water contact angles of composite membranes with varying PEG Mw.

**Figure 9 membranes-13-00378-f009:**
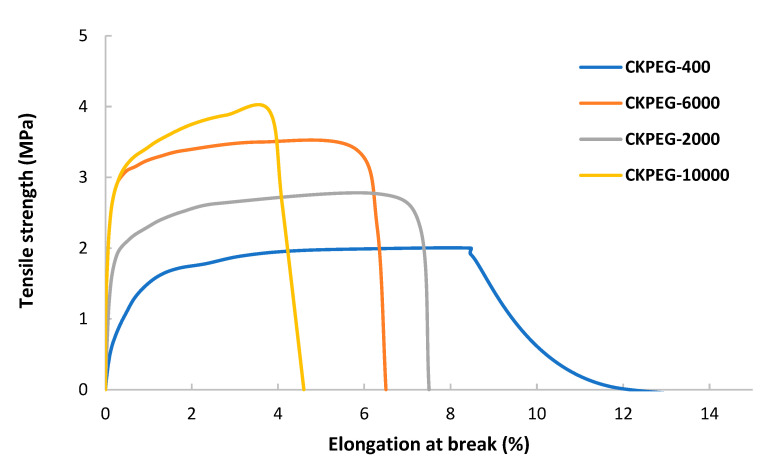
Mechanical properties of composite membranes with varying PEG Mw.

**Figure 10 membranes-13-00378-f010:**
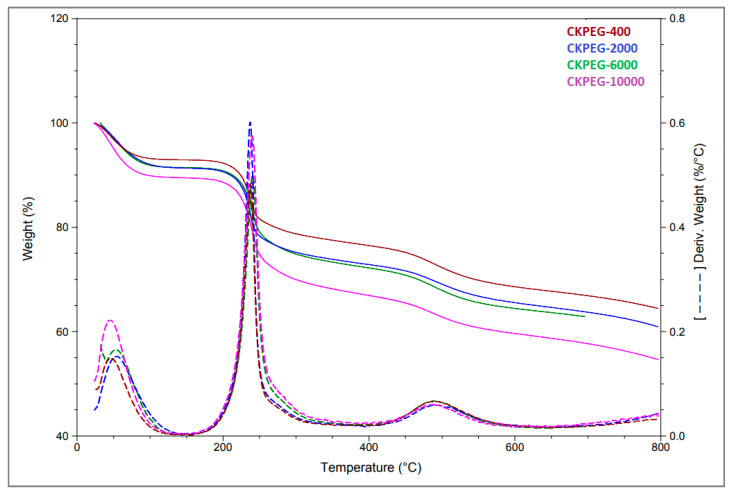
TG and DTG curves of composite membranes using varying PEG Mw.

**Figure 11 membranes-13-00378-f011:**
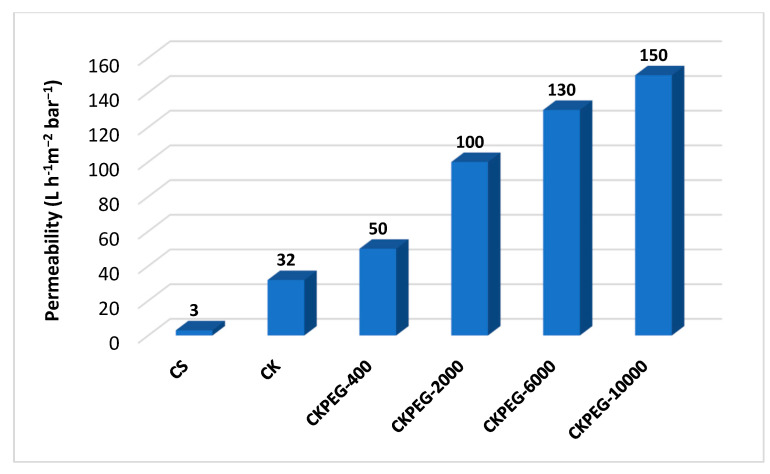
Pure water permeability values of the prepared membranes.

**Table 1 membranes-13-00378-t001:** Composition of the prepared dope mixtures.

Membranes	Component Part (*w*/*v*%) in 0.1 M Acetic Acid					
	CS	KO	PEG 400	PEG 2000	PEG 6000	PEG 10000
CS	4	-	-	-	-	-
CK	4	5	-	-	-	-
CKPEG-400	4	5	1	-	-	-
CKPEG-2000	4	5	-	1	-	-
CKPEG-6000	4	5	-	-	1	-
CKPEG-10000	4	5	-	-	-	1

**Table 2 membranes-13-00378-t002:** Surface pore size and porosity for CK membranes prepared using PEG of varying Mw as pore formers.

Membrane	Method	CKPEG-400	CKPEG-2000	CKPEG-6000	CKPEG-10000
Average surface pore diameter (nm)	SEM image analysis	50 ± 6	59 ± 7	71 ± 7	71 ± 3
	Nitrogen sorption–desorption	-	38	44	45
Surface porosity (%)	SEM image analysis	5 ± 2	9 ± 4	18 ± 4	17 ± 3
Pore volume (cm^3^·g^−1^)	Nitrogen sorption–desorption	-	0.0031	0.0028	0.0026

**Table 3 membranes-13-00378-t003:** Mechanical properties of the obtained composite membranes.

Sample Name	Tensile Strength (MPa)	Elongation at Break (%)
CK-PEG-400	2 ± 0.03	8.2 ± 0.5
CK-PEG-2000	2.8 ± 0.1	6.0 ± 0.1
CK-PEG-6000	3.5 ± 0.3	5.8 ± 0.1
CK5-PEG-10000	4 ± 0.3	4 ± 0.1

**Table 4 membranes-13-00378-t004:** Aqueous stability of CS and composite membranes using varying PEG Mw.

	CS	CK	CKPEG-400	CKPEG-2000	CKPEG-6000	CKPEG-10000
pH = 9	Insoluble	Insoluble	Insoluble	Insoluble	Insoluble	Insoluble
pH = 6.2	Partially soluble	Insoluble	Insoluble	Insoluble	Insoluble	Insoluble
pH = 4	Soluble	Partially soluble	Insoluble	Insoluble	Insoluble	Insoluble
pH = 2	Soluble	Soluble	Insoluble	Insoluble	Insoluble	Insoluble

**Table 5 membranes-13-00378-t005:** Pure water permeability comparison of CS-containing composite membranes prepared under different conditions.

Membrane Type	Preparation Method	Structure	Pore Size(nm)	PWP ^a^(L·h^−1^·m^−2^·bar^−1^)	Reference
MF/UF	Coating	CS/α-Al_2_O_3_	115	11	[43]
MF/UF	Coating	GA ^b^ crosslinked CS/clay ceramic support	36181	44900	[44]
UF	Blending	GA ^b^ crosslinked CS/CNC ^c^	13–10	6.4	[45]
NF	Blending	CS/GO ^d^	n.d.	0.58–1.31	[46]
MF	Blending	CS/PEG6000/MWCNT ^c^/iodine	n.d.	105	[47]
NF	Layer by layer	CS/GO ^d^ on PVDF UF membrane	n.d.	1–2.5	[48]
UF	Coating	GA ^b^ crosslinked CS/KO	16	11	[49]
UF	Blending	STPP crosslinked CS/PEG10000/KO	71	150	This work

^a^: Dead-end filtration; ^b^: glutaraldehyde; ^c^: carbon nanotube; ^d^: graphene oxide.

## Data Availability

The data presented in this study can be requested from the corresponding author for a reasonable reason.

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
