# Peer review of "Green Fabrication of Sustainable Porous Chitosan/Kaolin Composite Membranes Using Polyethylene Glycol as a Porogen: Membrane Morphology and Properties"

_membranes, 2023, doi:10.3390/membranes13040378_

Round 1

Reviewer 1 Report

The paper reports the green fabrication of sustainable porous chitosan/kaolin composite membranes using polyethylene glycol as a porogen, which is a topic of great interest to the researchers in the related areas. However, the paper needs improvement before acceptance for publication. My detailed comments are as follows:

1. The characterization and discussion on surface and bulk properties such as chemical structure and surface roughness should be supplemented to clarify the the role of KO and PEG in membrane formation.

2. It is more meaningful to use water solution with pollutants for filtration tests. And the anti-fouling property is suggested to be added.

3. Pore size and distribution is suggested to be measured via more accurate measurement tools.

4. More discussion on the role of KO on membrane formation and separation performance should be provided.

5. The comparison of separation performance with those in literatures should be provided to highlight the advantages of membrane prepared in this work.

Author Response

Indications and modifications in the text in response to the reviewer#1 are highlighted in yellow

The paper reports the green fabrication of sustainable porous chitosan/kaolin composite membranes using polyethylene glycol as a porogen, which is a topic of great interest to the researchers in the related areas. However, the paper needs improvement before acceptance for publication. My detailed comments are as follows:

  1. COMMENT: The characterization and discussion on surface and bulk properties such as chemical structure and surface roughness should be supplemented to clarify the the role of KO and PEG in membrane formation.

ANSWER:  

The surface of the membrane that is in direct contact with the feed plays a fundamental role in separation properties. We agree with the reviewer that discussion on surface characterization needs to be reorganized to clarify the role of KO and PEG in the surface properties of prepared membranes.

The ATR-FTIR technique enables the surface chemistry of membranes to be characterized over a maximum thickness of the micron range. Section “3.4. Surface properties of CS/KO membranes” was reorganized to first present the surface chemistry with a more detailed discussion of the spectra in the new Figure 7, followed by a paragraph on correlation of WCA values with surface chemistry and roughness.

“The surface of the membrane, which is in direct contact with the feed solution, plays a fundamental role in separation properties. Surface chemistry and wettability of the prepared CS/KO membranes were characterized using ATR-FTIR and WCA measurements.

Figure 7 presents the FTIR spectra of the CKPEG-10000 membrane surface compared to that of PEG-10000 powder and CK membrane. The CK spectrum showed the presence of KO by its very strong bands at 996, 1025, and 1112 cm−1 ascribed to the stretching vibrations of Si–O and of CS at 1650 and 1590 cm−1 attributed to C=O stretching vibration (amide I band) and N-H stretching vibration (amino II band), respectively. The interaction between CS and KO was evidenced by a shift of these characteristic bands compared to the starting materials [19,37]. This effect confirmed the ionic exchange reaction occurring between CS and kaolin-clays and intercalation into the KO structure [19]. PEG is characterized by two broad bands at about 2974–2881 and 1148-1060 cm-1, which are assigned to the absorption of aliphatic C-H and ether C-O-C groups, respectively [38]. Using PEG as an additive in the CK composite membrane, notable changes appeared in the 1150-950 cm-1 range, which can be explained by the signature of the PEG ether groups. This result confirmed the presence of trapped PEG chains at the membrane surface.

Hydrophilicity is a desirable property for filtration membranes in aqueous media because it is expected to give high permeate flow rates and reduce fouling. Figure 8 shows the results of WCA measurements for the different composite membranes prepared. Wettability is a function of surface chemistry but also of its roughness [38]. The pure CS membrane showed a WCA value of about 57°, before the water droplet was fully absorbed by the matrix. This result reflected the marked hydrophilic character of the CS polysaccharide backbone and the smooth membrane surface, as observed by SEM (Figure 5). It can be clearly seen that the surface morphology after KO loading becomes rougher compared to that of the pure CS membrane. In this case, the surface roughness plays a key role in imparting an apparent hydrophobic character to the membrane surface. Aggregated particles can be observed uniformly distributed on the CK membrane surface inducing a high degree of surface roughness explaining the higher WCA value. On the other hand, the H and ionic bonds between the functional groups of CS and KO generated a stiffer membrane surface and reduced interactions of hydrophilic groups with water molecules, which also participated in the increase of the WCA value [19]. In contrast, the incorporation of PEG caused drastic changes in the hydrophilic properties of the resulting membranes. The PEG addition induced a smoothening effect on the membrane surface, which was particularly noticeable in the case of CKPEG-10000. The WCA of the CKPEG-10000 showed a value as low as about 62° (Figure 8), indicating a significant increase in wettability with respect to the CK membrane surface. Further, ATR-FTIR analysis showed that a part of PEG chains remained entangled in the CS/KO network, leading to a proportion of residual PEG on the membrane surface. Therefore, it was assumed that the highly hydrophilic nature of the PEG chains located at the interface resulted in an increase of wettability through greater affinity of the membrane surface for water molecules [31,39].

  1. COMMENT: It is more meaningful to use water solution with pollutants for filtration tests. And the anti-fouling property is suggested to be added.

ANSWER:

This work focused on the role played by a pore former on the structure and morphology of CS/KO composite membrane.  Therefore, the main objective was the characterization of the membranes in terms of physical properties in relation to the obtained morphology. The pure water permeability was determined in order to highlight the improvement of porosity allowed by the addition of PEG.

We agree with the reviewer that filtration tests under real conditions would provide a better insight into the application properties of these membranes. However, this point is beyond the scope of the present work and will be presented in future papers as indicated in conclusion section “these composite membranes could be a promising alternative to address the sustainability challenge in membrane fabrication for water treatment. Future work will investigate the potential of these membranes for the filtration of real feed solutions.”

  1. COMMENT: Pore size and distribution is suggested to be measured via more accurate measurement tools.

ANSWER:

SEM image analysis of polymer membrane surfaces is a rapid method for dertermining pore size and distribution that does not require specific equipment. It has been proven to give results in good agreement with other techniques like rejection of model molecules and liquid-liquid porometry [29]. Because it is a local measurement, analysis has to be carried out in several images of different samples. It is a well-established technique for polymeric membranes (see for exemple [31]). The references mentioned above were added in the text (page 9, line 337).

Other methods for determining pore size distribution exist, such as mercury intrusion porosimetry and nitrogen adsorption techniques (BET, BJH...). However, the authors would like to make the following comments on the limitations of these techniques:

  • Mercury intrusion porometry is a well-suited technique for measuring pore diameter in mechanically stable membranes such as ceramic materials. For example, a pressure above 3.5 MPa is required for mercury to penetrate pores smaller than 0.5 µm, so polymer-containing membranes are generally not robust enough to withstand such pressures without collapsing (see table 3).
  • Nitrogen adsorption techniques are more suitable for the analysis of the mesoporous domain. In this case, capillary condensation at 77 K is generally accepted as responsible for pore filling. Two remarks should be made. Again, the calculation requires that the pores be rigid and undeformable, which is generally not possible with polymer-containing membranes, because vacuum and heating steps are performed before the measurement. Second, the volume of nitrogen adsorbed by the top layer is very small, so it is generally impossible to determine the surface pore size of an asymmetric polymer-containing membrane.

To illustrate these comments, Table 2 was completed by the results obtained by BET measurements and the following sentences were added.

“The data obtained from SEM image analysis were compared in Table 2 with the pore size and volume determined by BET measurements. The pore size is systemically smaller and the pore volume shows an opposite trend to that observed in the SEM images (Figures 3 and 5). The procedure used in the BET method probably led to a partial collapse of the membrane structure the larger the PEG Mw and, hence, the higher the porosity. Therefore, the SEM image analysis appeared to reflect more accurately the morphology of CS/KO composite membranes.”

Table 2. Surface pore size and porosity for CK membranes prepared using PEG of varying Mw as pore former.

Membrane

Method

CKPEG-400

CKPEG-2000

CKPEG-6000

CKPEG-10000

Average surface pore diameter (nm)

SEM image analysis

50±6

59±7

71±7

71±3

Nitrogen sorption-desorption

-

40

44

50

Surface porosity (%)

SEM image analysis

5±2

9±4

18±4

17±3

Pore volume (cm-3.g-1 )

Nitrogen sorption-desorption

-

0.0031

0.0028

0.0026

  1. COMMENT: More discussion on the role of KO on membrane formation and separation performance should be provided.

ANSWER:

The role played by KO in membrane bulk formation has been explained in ref. 19 as indicated in section, “3.2. Membrane morphology » lines 260-263.

“The cross section of the CS membrane (Figure 3a) showed a dense structure without any holes while that of the CK (figure 3b) membrane had a loose-layered structure that was assumed to be due to the intercalation of CS chains between KO sheets [19]”

To clarify, the appearance of the surface porosity related to the introduction of KO particles, a zommed-in SEM image of the CK membrane surface was inserted in Figure 5 and discussed in Section “3.3. Surface porosity, average pore size and pore size distribution” lines 322-331.

“Figure 5 shows the evolution of the surface morphology of the pure chitosan (CS) film and the CK composite membrane. The CS films have a continuous structure with a smooth, homogeneous and compact surface without pores or cracks, while the CK composite membrane has a much rougher surface morphology. A zoomed-in image clearly reveals the presence of tiny pores inside the particles, whose size can be estimated between 20 and 60 nm by image analysis. The origin of the pore structure was assumed to come from the migration of chitosan molecules to the negative charges between the clay sheets. Thus, the penetration and intercalation of chitosan chains into the clay layers would result in the formation of a three-dimensional network of nano-sized pores in the composite membrane [19].”

The two following sentences are also added in section “3.6. Membrane permeability”.

“The CS membrane exhibited very poor permeability due to its dense structure and lack of porosity (Figure 3 and 5). The addition of KO particles led to an improvement of one-order of magnitude in PWP, which is consistent with the porosity revealed by SEM (Figure 5).”

  1. COMMENT: The comparison of separation performance with those in literatures should be provided to highlight the advantages of membrane prepared in this work.

ANSWER:

The following discussion illustrated by Table 5 was added in the section “3.6. Membrane permeability”.

“Many works have investigated CS-containing composites for various applications such as membrane filtration, metal ions, and adsorption of dyes and pollutants [40]. Table 5 summarizes for comparison ceramic and polymer composites prepared according to several methods including blending (as in this work), coating or layer-by-layer deposition on a pre-formed membrane. CS was immobilized either by chemical crosslinking with glutaraldehyde (GA) or hydrogen bonding and ionic interactions with negatively charged component (ceramic and carbon derivatives). Although the filtration range varies from MF to NF, it can be seen that the filtration performance of the CS/KO composite membranes prepared in this work using PEG Mw of 10,000 g.mol-1 compares favorably with that of other works. This is probably due to their asymmetric morphology with a relatively thin filtration layer on a substructure providing very low resistance permeate flow.” 

Table 5. Aqueous stability of CS and composite membranes using varying PEG Mw.

Membrane type

Preparation method

Structure

Pore size

(nm)

PWPa

(L.h-1.m−2.bar−1)

Reference

MF/UF

Coating

CS / α-Al2O3

115

11

[41]

MF/UF

Coating

GAb crosslinked CS / clay ceramic support

36

181

44

900

[42]

UF

Blending

GAb crosslinked CS/CNCc

13-10

6.4

[43]

NF

Blending

CS/GOd

n.d.

0.58-1.31

[44]

MF

Blending

CS/PEG6000/MWCNTc/Iodine

n.d.

105

[45]

NF

Layer by layer

CS/GOd on PVDF UF membrane

n.d.

1-2.5

[46]

UF

Coating

GAb crosslinked CS / KO

16

11

[47]

UF

Blending

STPP crosslinked CS/PEG10000/KO

71

150

This work

  1. Dead-end filtration; b. Glutaraldehyde; c. Carbon nanotube; d. Graphene oxide.

Reviewer 2 Report

This work reported the synthesis, characterization, and properties of porous chitosan/kaolin composite membranes using PEG as a pore former additive. Although the physical and chemical properties of the composite membrane have been described clearly, its formation mechanism considering PEG ought to be illustrated thoroughly; otherwise, the essence of this paper is lacking. In addition, several details should be added; thus, the reviewer thinks that this manuscript can be considered after major revision.

1)Should provide zeta potential of the casting mixture.

2)Surface porosity of the composite membrane should be characterized via BET measurement.

3)Could the author explains how PEG does not influence the chemical stability of the composite membrane?

4) Permeability should be correlated with molecular weight cut-off (MWCO).  

5)To support the author's rationale for sustainable membranes, a retention expansion should be added.

6)Need to show tensile-elongation curve to describe well its mechanical properties. 

Author Response

Indications and modifications in the text in response to the reviewer#2 are highlighted in green

This work reported the synthesis, characterization, and properties of porous chitosan/kaolin composite membranes using PEG as a pore former additive. Although the physical and chemical properties of the composite membrane have been described clearly, its formation mechanism considering PEG ought to be illustrated thoroughly; otherwise, the essence of this paper is lacking. In addition, several details should be added; thus, the reviewer thinks that this manuscript can be considered after major revision.

  1. COMMENT: Should provide zeta potential of the casting mixture.

ANSWER:

The following paragraph was added to the beginning of “3.2. Membrane morphology” section to clarify the role of PEG in membrane formation from the casting mixture.

“KO particles have a negative surface charge in the pH range 3-9 (isoelectric point about 3). CS is positively charged in the same pH range (pKa = 6.5). Mixing the two leads to strong interactions through charge neutralization of KO particles and hydrogen bonding [19] so that the zeta potential shifts to less negative values depending on the ratio CS/KO and the CS Mw [32]. The zeta potential of CS/KO suspensions prepared at pH 4 in this work was positive and equal to about 15-20 mV. It has been reported that PEG can also adsorb to KO particles and cause an increase in zeta potential although these molecules are neutral [33]. However, due to the large difference in concentration, Mw and interaction strength compared to CS, PEG probably decorated the CS/KO surface as no significant difference in zeta potential was observed. Upon neutralization of the film forming system, precipitation-flocculation occurred. It was then expected that PEG would be released into the coagulation bath and act as a pore former.”

  1. COMMENT: Surface porosity of the composite membrane should be characterized via BET measurement.

ANSWER:

Nitrogen adsorption techniques are suitable for the analysis of the mesoporous domain. In this case, capillary condensation at 77 K is generally accepted as responsible for pore filling. Two remarks should be made. The calculation requires that the pores be rigid and undeformable, which is generally not possible with polymer-containing membranes, because vacuum and heating steps are performed before the measurement. Second, the volume of nitrogen adsorbed by the top layer is very small, so it is generally impossible to determine the surface pore size of an asymmetric polymer-containing membrane. To illustrate these comments, Table 2 was completed by the results obtained by BET measurements and the following sentences were added.

“The data obtained from SEM image analysis were compared in Table 2 with the pore size and volume determined by nitrogen sorption-desorption measurements. The pore size is consistently smaller and the pore volume shows an opposite trend to that observed in the SEM images (Figures 3 and 5). The procedure used in the BET method probably led to a partial collapse of the membrane structure the larger the PEG Mw and, hence, the higher the porosity. Therefore, the SEM image analysis appeared to reflect more accurately the morphology of CS/KO composite membranes.”

Table 2. Surface pore size and porosity for CK membranes prepared using PEG of varying Mw as pore former.

Membrane

Method

CKPEG-400

CKPEG-2000

CKPEG-6000

CKPEG-10000

Average surface pore diameter (nm)

SEM image analysis

50±6

59±7

71±7

71±3

Nitrogen sorption-desorption

-

38

44

45

Surface porosity (%)

SEM image analysis

5±2

9±4

18±4

17±3

Pore volume (cm3.g-1 )

Nitrogen sorption-desorption

-

0.0031

0.0028

0.0026

  1. COMMENT: Could the author explains how PEG does not influence the chemical stability of the composite membrane?

ANSWER:

As pointed out in the manuscript, pure CS membranes have some shortcomings in terms of mechanical strength, thermal stability and solubility in acidic media. As an example, CS can easily dissolve partially in water and completely in acidic media, which negatively affects the properties of membranes applicable to water treatment. Several strategies to overcome these limitations can be found in the literature, such as the preparation of composite membranes with mixed CS/inorganic particles and/or a cross-linking of CS. The CS/KO composite membranes prepared in this work were ionically crosslinked with STPP (section 2.2.) which stabilized them up to pH 2 [19-21]. PEG was introduced into the formulation as a pore former and most of it was released into the coagulation bath. Therefore, PEG does not affect the chemical stability but rather the mechanical properties due to the increase of the pore volume as shown in Table 3.

  1. COMMENT: Permeability should be correlated with molecular weight cut-off (MWCO).  

ANSWER:  

MWCO is generally related to the pore size in UF membranes, aside the hydrodynamic and adsorption phenomena associated with operating conditions and specific compounds in feed. In the other hand, permeability is not directly related to MWCO, as surface wettability and membrane morphology, including thickness of the filtration layer, pore tortuosity and surface density also have a strong influence on it.

Thus, permeability was correlated with membrane morphology and hydrophilicity. In particular, it was highlighted that straight finger-like substructure counteracted the increase in filtration thickness in CKPEG-10000 compared to CKPEG-6000.

  1. COMMENT: To support the author's rationale for sustainable membranes, a retention expansion should be added.

ANSWER:

This work focused on the role played by a pore former on the structure and morphology of CS/KO composite membrane.  Therefore, the main objective was the characterization of the membranes in terms of physical properties in relation to the obtained morphology. The pure water permeability was determined in order to highlight the improvement of porosity allowed by the addition of PEG.

We agree with the reviewer that filtration tests under real conditions would provide a better insight into the application properties of these sustainable membranes. However, this point is beyond the scope of the present work and will be presented in future papers as indicated in conclusion section “these composite membranes could be a promising alternative to address the sustainability challenge in membrane fabrication for water treatment. Future work will investigate the potential of these membranes for the filtration of real feed solutions.”

  1. COMMENT: Need to show tensile-elongation curve to describe well its mechanical properties. 

ANSWER:

A figure was added as suggested.

Figure 9. Mechanical properties of composite membranes with varying PEG Mw.

Reviewer 3 Report

             The manuscript investigate CS/KO composite membranes prepared using PEG as a pore forming additive. The work is within the scope of the journal and the major comments are as follows: 

1、To evaluate the effect of PEG additives on membrane structure, the authors refer to the change in the mean pore size of the membrane in the manuscript. If possible, please also provide the porosity variation of the corresponding membrane to complete the description.

2、The pore size of the membrane is not small and the contact angle is also low, why is the flux not good?

3、The stability of the membranes should be tested.

Author Response

Indications and modifications in the text in response to the reviewer#3 are highlighted in blue

The manuscript investigate CS/KO composite membranes prepared using PEG as a pore forming additive. The work is within the scope of the journal and the major comments are as follows: 

  1. COMMENT: To evaluate the effect of PEG additives on membrane structure, the authors refer to the change in the mean pore size of the membrane in the manuscript. If possible, please also provide the porosity variation of the corresponding membrane to complete the description.

ANSWER:

Table 2 was completed to provide the surface porosity variation as void percentage determined by SEM image analysis and pore volume by BET.

Table 2. Surface pore size and porosity for CK membranes prepared using PEG of varying Mw as pore former.

Membrane

Method

CKPEG-400

CKPEG-2000

CKPEG-6000

CKPEG-10000

Average surface pore diameter (nm)

SEM image analysis

50±6

59±7

71±7

71±3

Nitrogen sorption-desorption

-

38

44

45

Surface porosity (%)

SEM image analysis

5±2

9±4

18±4

17±3

Pore volume (cm3.g-1 )

Nitrogen sorption-desorption

-

0.0031

0.0028

0.0026

  1. COMMENT: The pore size of the membrane is not small and the contact angle is also low, why is the flux not good?

ANSWER:

We agree with the reviewer. The following paragraph was added in “Membrane permeability” section.

As a conclusion, the characteristics of CS/KO composite membranes in terms of pore size and contact angle should lead to better permeability. However, membrane permeability depends on the surface wettability and pore size, but also on other factors such as filtration layer thickness, surface pore density and pore tortuosity. For example, the CKPEG-10000 membrane has a top filtration layer of about 25 µm thickness that strongly hindered the permeate flow (Figures 3f and 4). It is obvious that a thinner thickness would lead to better permeability. This point should be optimized to obtain more efficient membranes.

  1. COMMENT: The stability of the membranes should be tested.

ANSWER:

The stability of CS/KO composite membranes were tested in terms of mechanical (Figure 9 and Table 3), thermal (Figure 10) properties and chemical stability against alkaline and acidic dissolution in the pH range 2-9 (Table 4).

Round 2

Reviewer 1 Report

The work can be accepted.

Reviewer 2 Report

The authors have addressed the comments by reviewer pretty well. Some missing data have also been added accordingly. 

So, I can reccomend the present state of the manuscript for publication.